# Design of a BIST implemented AES crypto-processor ASIC

**Md. Liakot Ali**[1][*], **Md. Shazzatur Rahman**[1], **Fakir Sharif Hossain**[2]

**1** Institute of Information and Communication Technology, Bangladesh University of Engineering and Technology, Dhaka, Bangladesh, **2** Department of Electrical and Electronic Engineering, Ahsanullah University of Science and Technology, Dhaka, Bangladesh

☯ These authors contributed equally to this work.
* liakot@iict.buet.ac.bd

**Data Availability Statement:** All data are within the manuscript and its Supporting information files.

**Funding:** The author(s) received no specific funding for this work.

## Abstract

This paper presents the design of a Built-in-self-Test (BIST) implemented Advanced Encryption Standard (AES) cryptoprocessor Application Specific Integrated Circuit (ASIC). AES has been proved as the strongest symmetric encryption algorithm declared by USA Govt. and it outperforms all other existing cryptographic algorithms. Its hardware implementation offers much higher speed and physical security than that of its software implementation. Due to this reason, a number of AES cryptoprocessor ASIC have been presented in the literature, but the problem of testability in the complex AES chip is not addressed yet. This research introduces a solution to the problem for the AES cryptoprocessor ASIC implementing mixed-mode BIST technique, a hybrid of pseudo-random and deterministic techniques. The BIST implemented ASIC is designed using IEEE industry standard Hardware Description Language(HDL). It has been simulated using Electronic Design Automation (EDA)tools for verification and validation using the input-output data from the National Institute of Standard and Technology (NIST) of the USA Govt. The simulation results show that the design is working as per desired functionalities in different modes of operation of the ASIC. The current research is compared with those of other researchers, and it shows that it is unique in terms of BIST implementation into the ASIC chip.

## 1 Introduction

Information and Communication Technology (ICT) has become an integral part of our daily life. People want to protect their information from unauthorized access and data corruption. Cryptography plays an important role in protecting and securing information during communication [1]. There are two types of cryptographic algorithms: symmetric and asymmetric. Asymmetric cryptography is more secured than symmetric cryptography, but it is complex and uses much longer keys which in turn makes it slower and not industry-accepted, and not suitable for bulk data transmission than its counterpart symmetric cryptography. However, research is going on for the improvement of the performance of the asymmetric cryptography algorithm [2–6]. On the other hand, symmetric cryptography is less complex and executes

**Competing interests:** The authors have declared that no competing interests exist.

faster than asymmetric cryptography, and so it has been accepted as an industry-standard cryptographic algorithm and is being widely used to secure sensitive, secret, or classified information in government, healthcare, banking, and other industries. Previously researchers proposed plenty of symmetric cryptographic algorithms [7–11]. Advanced Encryption Standard (AES) is one of the most secured and chosen US NIST of USA Govt for its military among the multiple cryptographic algorithms. Crypt-analytical attacks such as Brute-force, Linear crypt-analysis, Differential crypt-analysis, etc., have been proven ineffective in breaking the AES. It has been shown that even with a supercomputer, it would take 1 billion years (which is more than the age of the universe: 13.75 billion years) to crack this algorithm [12]. Due to this impressive security potentiality of AES, it is being used in various emerging applications, either in software or hardware implementations. Hardware implementation of the algorithm offers higher security and speed than that of its software implementation. Due to enormous speed and security performances, now a lot of research for hardware realization of the AES crypto-processor is reported in the literature [13–21]. Some of the research focuses on hardware resource optimization [13–15], while some other on speed optimization [16–18] and some other on power consumption optimization [19–21].

Hardware implementation can be realized using two approaches: (i) Field Programmable Gate Array (FPGA) technology (ii) Application Specific Integrated Circuit (ASIC) technology. ASIC is a VLSI chip with an optimized circuit for a particular application, whereas FPGA is also an off-the-shelf VLSI chip manufactured for general purposes. ASIC is much more efficient in terms of power, area, and speed than FPGA [22].

In today's world, for many emerging and demanding applications where security and reliability are an issue, ASIC implementation of AES cryptoprocessor is the best solution for the clients. However, for designing ASIC or any complex chip, Design for Testability (DFT) is a prime concern because testing a VLSI chip using Automatic Test Equipment (ATE) is highly complex, time-consuming as well as expensive [23, 24]. To deal with the testing problem at the chip level, incorporating Built-in Self-Test (BIST) capability inside a chip is a widely accepted approach [25–30] and it is a norm of this day in the VLSI industry. BIST is a mode of operation of a chip other than its normal mode, where when a chip is switched to this mode, it performs its test by itself. Nowadays mixed-mode testing approach [31–35] is the state of the art technique for BIST implementation where the ASIC is tested using both pseudo-random test patterns for easy to test faults and deterministic test patterns for hard to detect faults, and thereby, maximum fault coverage is achieved. In this research, the mixed-mode BIST technique has been incorporated into the design of the AES cryptoprocessor ASIC.

The main contributions of this work are summarized as below:

- To introduce the concept of mixed-mode BIST technique in designing the AES Crypto-processor ASIC chip. The idea is a unique but significant feature of the AES crypto-processor ASIC, which is not reported in the literature yet.

- To implement the mixed-mode BIST technique in the design of the ASIC and simulate the design to ensure the functionalities in a different mode of operation using the ModelSim EDA tool.

- To validate the simulation results using Quartus II EDA tool and find the resource (logic gates) requirements for the desired functionalities.

The remaining sections of this paper are organized as follows. In Section-2, a discussion on the AES algorithm and its different levels of functioning are presented. Section-3 describes the evolution of BIST towards the mixed-modeBIST with description. The architecture of the proposed AES crypto-processor ASIC BIST is presented in Section-4. Experimental results are

shown in Section-5. A comparison to other works is presented in Section-6. Finally, we conclude the work in Section-7.

## 2 AES algorithm

AES accomplishes all its operations on bytes rather than bits. AES interprets a plaintext block of 128 bits as 16 bytes. A 4x4 matrix is used to represent these 16 bytes. Fig 1 shows the overall operational structure of the AES algorithm. The number of rounds based on which the AES

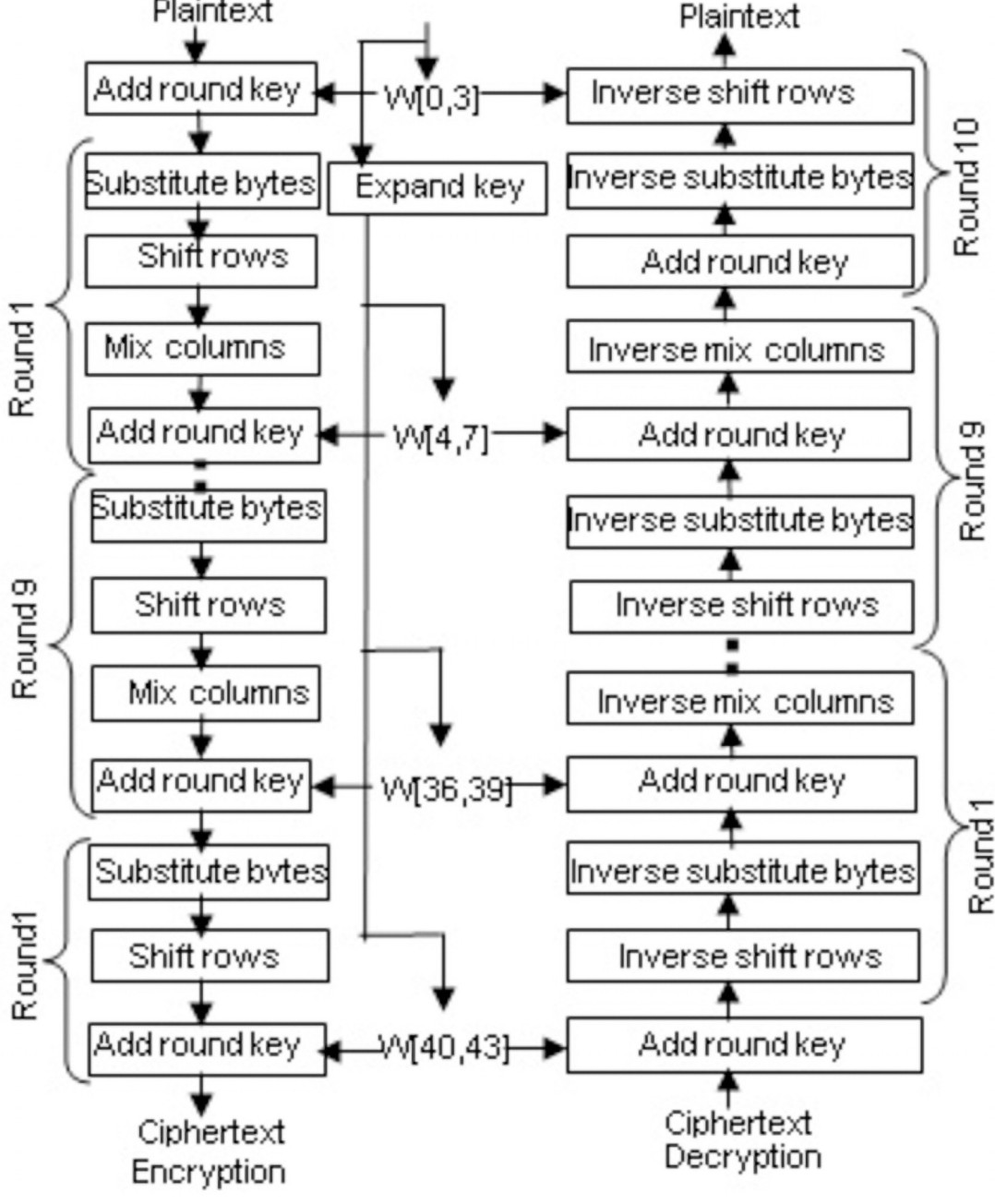

**Fig 1. AES encryption and decryption.**

algorithm performs encryption or decryption is variable and depends on the key length. In AES for rounds, 10, 12, and 14 the key lengths are 128 bits, 192 bits, and 256 bits, respectively. In this research, we focused only on the key length of 128 bits. For encryption, each round consists of four sub-operations. The sub-operations are as follows:

Byte substitution: AES defines a fixed table called s-box consists of 256 values for the substitution. AES works using the 16 bytes of the state array, employs each byte as an index into the 256-byte substitution table, and substitutes the byte with the value from the substitution table.

Shift rows: In this sub-operation, each of the four rows of the state matrix is shifted to the left. The first row remains unchanged. The second row is shifted one-byte position to the left. The third row is shifted two-byte positions to the left. The fourth row is shifted three-byte positions to the left.

Mix columns: Mix columns sub-operation performs a matrix multiplication of a given state matrix with a static matrix. This sub-operation is treated as the primary source of diffusion. Finally, we obtain a new state matrix containing 16 new bytes. Add round key: The state matrix containing 16 bytes is considered 128 bits and is XORed to the 128 bits of the round key. If this is the last round, then the output is our expected ciphertext; otherwise, the resulting 128 bits are interpreted as 16 bytes and are used in the next round.

In AES, the four sub-operations can be inverted because AES is not a Feistel network. The inverse sub-operations for each round in the decryption process are shown in Fig 1. We know that the XOR operation is its own inverse. The add round key sub-operation is performed in the same manner for both encryption and decryption processes.

Inverse mix columns: Inverse mix columns sub-operation is just the reverse of the mix columns sub-operation. Here the static matrix used for the matrix multiplication is the reverse of that of the mix columns sub-operation.

Inverse shift rows: In this sub-operation, each of the four rows of the state matrix is shifted to the right. The first row remains unchanged. The second row is shifted one-byte position to the right. The third row is shifted two-byte positions to the right. The fourth row is shifted three-byte positions to the right.

Inverse sub bytes: AES S-Box follows a one-to-one mapping; an inverse table of the fixed S-Box table is used. Similarly, each byte of the state array is used as an index into the 256-byte substitution table and substitutes the byte with the value from the substitution table.

Key expansion: Key expansion algorithm generates the round keys by employing the initial key for both the encryption and decryption processes. The four-word key is feed to the AES key expansion algorithm, and eventually, we get a linear array of 44 words. Each sub-key is 128 bits in length. The initial is held into the first four words of the expanded key. The preceding word and the word four positions back are XORed to generate each added word. The whole process consists of two sub-functions called rotation word and substitution word. In the first sub-function, a one-byte left shift is performed on a word, and in the second sub-function, a byte substitution is performed on each byte of its input word, using the S-box.

## 3 VLSI testing towards mixed-mode BIST

IC manufacturers use ATE for testing IC. The deterministic testing approach introduced in 1960 is adopted in the ATE [36]. ATE cannot cope with the continuously increasing complexities of IC in today's semiconductor world. To overcome the serious drawbacks suffered by ATE such as high equipment cost, slow test speed, yield loss, etc. [37, 38] a new testing approach, Built-In Self-Test (BIST) technique, is proposed by researchers. Initially, the same deterministic approach as used in ATE was used in BIST. However, later on, a number of techniques have been proposed to improve the performance of the BIST such as pseudo-random

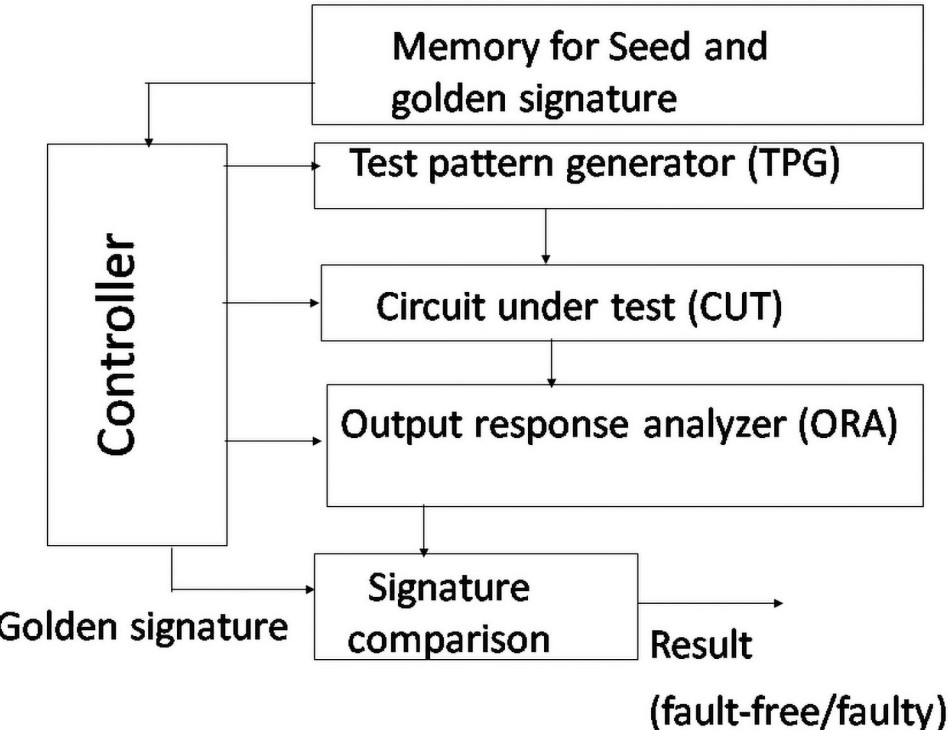

**Fig 2. AES BIST architecture.**

testing approach [39] weighted random testing approach [40, 41], mixed-mode testing approach [31–33] etc. Later on, different variants of mixed-mode testing approaches are proposed [34, 35]. The mixed-mode testing approach is a long-established and popular method for BIST where acceptable and higher fault coverage can be achieved using a lower number of test vectors in testing the VLSI chip. It is a hybrid test technique where a pseudo-random test technique follows the deterministic test technique. This approach exploits the advantages of both the pseudo-random test technique and the deterministic test technique. In this approach, testing is performed in two phases. In the first phase, a linear feedback shift register (LFSR) is employed to generate the required number of pseudo-random test vectors (PRV), which are then fed to the circuit under test (CUT), and around more than 80% easy to detect faults are possible to be detected. Then the remaining faults are detected employing deterministic test vectors. Deterministic test vectors are generated using the reseeding technique of LFSR, where the required numbers of seed values are stored in the memory. They are loaded into the LFSR in need and employed to generate the required number of deterministic test patterns. In this process, we can use the same LFSR to be used for generating PRVs. Fig 2 shows the mixed-mode BIST architecture. It consists of a controller, memory, Test Pattern Generator (TPG), Output Response Analyzer (ORA), etc. BIST controller reads the seed and initializes the LFSR for generating the test required deterministic vectors applied to the CUT, and the output is fed into ORA. Then signature generated into the ORA is compared with that of the golden signature, and the status of CUT is determined as faulty/fault-free.

## 4 Architecture of AES Crypto ASIC

Figs 3 and 4 show the functional block diagram of the AES crypto ASIC with BIST and its flowchart of operation. The ASIC comprises five prime modules. The encryption module,

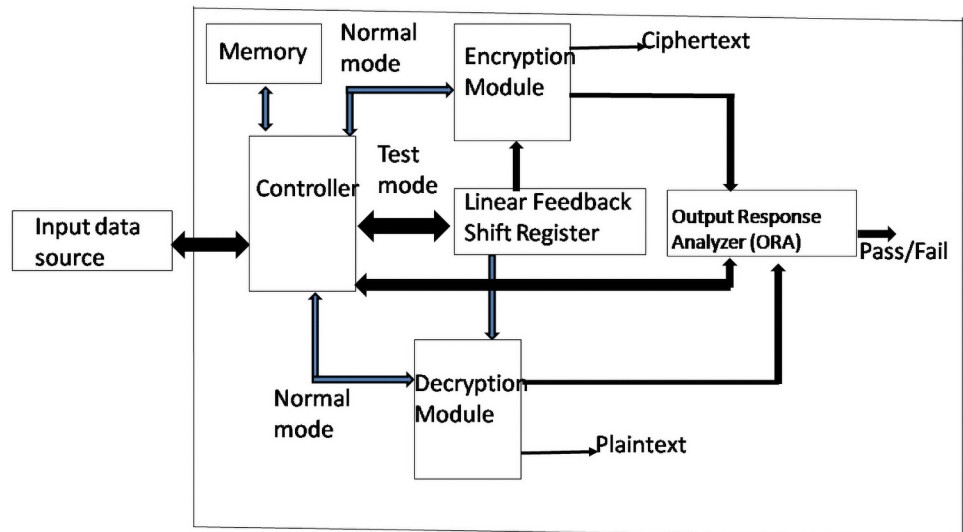

**Fig 3. Functional blocks of the AES Crypto ASIC with BIST.**

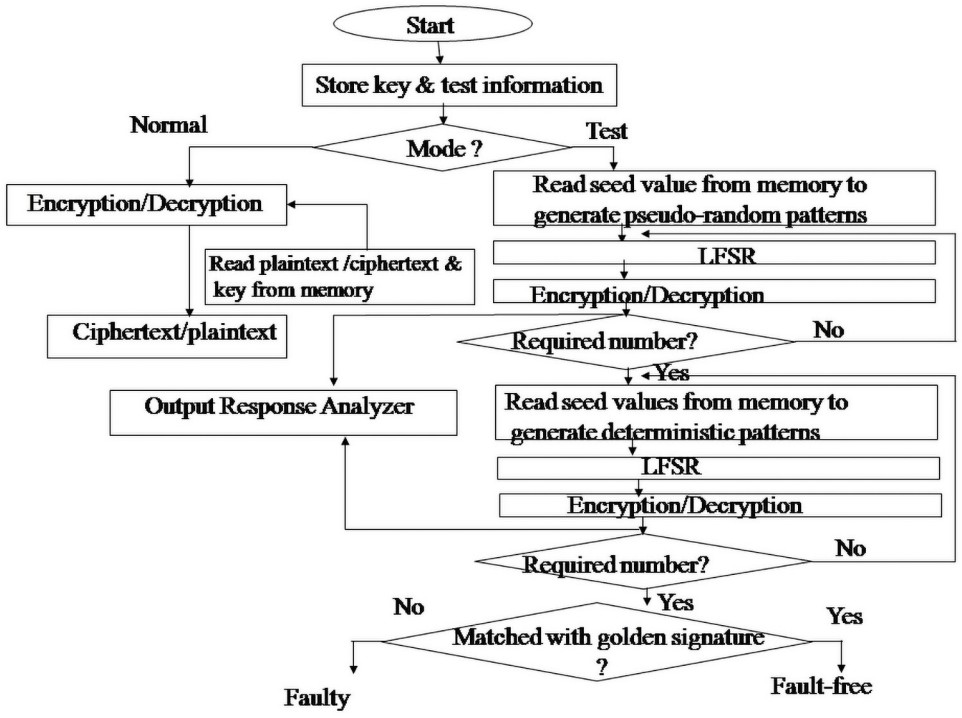

**Fig 4. Flowchart of the AES Crypto ASIC with BIST.**

decryption module, controller module, linear feedback shift register, and output response analyzer (ORA) module. The controller module is responsible for controlling the sequences in which other modules are to be activated. When the encryption module works in normal mode, then the required plaintext and the input key are read from the input data source. After ten rounds, the resultant ciphertext is obtained. All the inputs and the outputs are verified as per

NIST provided inputs and outputs. Similarly, in the case of decryption under normal mode, the required ciphertext is read from the input data source, and then after ten rounds, the resultant plaintext is displayed, which is also verified in the same way as provided by NIST. When the test mode is activated, whether the encryption module or decryption module is to be tested at first, the required number of pseudo-random test patterns generated from the linear feedback shift register (LFSR) is feed to the encryption or decryption module. When the required number is reached, then the test switches to the deterministic test patterns instead of pseudo-random test patterns.

The same LFSR is used to generate the required number of deterministic test patterns, and those are fed to the encryption or decryption module. All the responses obtained from the encryption or decryption are fed to the ORA module. The final output received from the ORA module, which can be thought of as our candidate signature, is compared with the golden signature stored in the memory. Our research has two golden keys, one for encryption and the other for decryption, stored in the memory. If there is a perfect match, that indicates a successful test, otherwise failure.

## 5 EDA simulation results and discussion

In this research, the BIST implemented AES cryptoprocessor ASIC is designed using Verilog HDL, and the functionalities of the ASIC are simulated using the Modelsim EDA simulator. The simulation results in different modes of operation using the NIST-provided inputs and outputs to ensure that the ASIC is working as per desired functionalities. The design of the ASIC is again compiled and simulated using the FPGA-based EDA tool Quartus II. The specific FPGA device is "EP2C70F896C6" from the Cyclone II family mounted on an educational Kit Altera DE2 which is available in the embedded system Lab of IICT, BUET. The specifications of the FPGA Device are supply voltage 1.15 V to 1.25 V, maximum operating frequency 260 MHz, total pins 622, number of logic elements 68416. The design of the ASIC is compiled and simulated using the EDA tool, and found the complete resources are shown in Table 1.

Fig 5 shows the simulation result of the encryption module in normal mode. Six input pins of the ASIC are to be used to control the encryption and decryption process either in normal mode or in test mode. The six input pins are "normalEncryption", "normalDecryption","bistMode", "encryptionForRandom", "decryptionForRandom", and "decryptionFollowsEncryption".

Among the six input pins, one input pin "normalEncryption" is set to one, and the other five input pins are set to zero so that the initial key"000102030405060708090a0b0c0d0e0f" and the plaintext "00112233445566778899aabbccddeeff" are loaded into the input pins "keyI" and "initialState" respectively as shown in Fig 5. The output pin "cipherTextValue" holds the desired ciphertext value "69 c4 e0 d8 6a 7b 4 30 d8 cd b7 80 70 b4 c5 5a". All the inputs and outputs are verified according to the NIST provided test inputs and outputs.

Fig 6 shows the simulation result of the decryption module in normal mode. Among the six input pins, one input pin "normalDecryption" is set to one and the other five input pins are set

**Table 1. Used resources in FPGA simulation.**

| Resources | Usage |
|---|---|
| Total logic elements | 36,507 |
| Total combinational functions | 36,081 |
| Dedicated logic registers | 9,323 |
| Total pins | 553 |

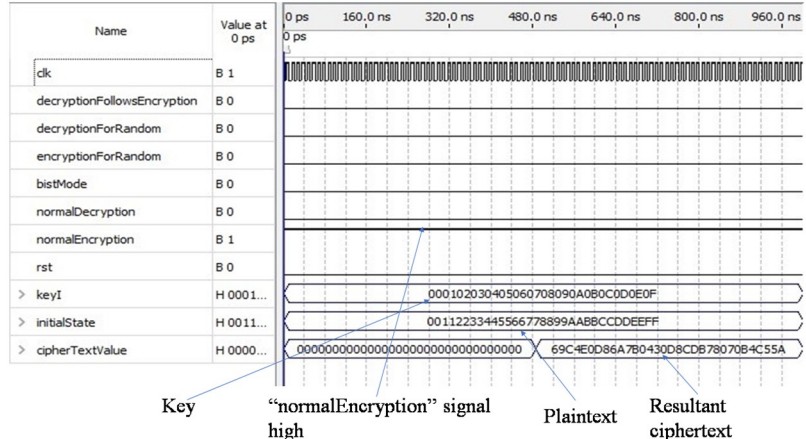

**Fig 5. Simulation result of encryption module in normal mode.**

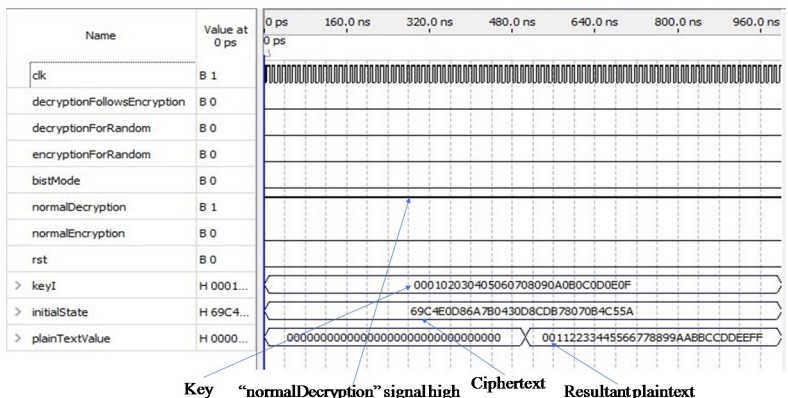

**Fig 6. Simulation result of decryption module in normal mode.**

to zero so that the initial key "00 01 02 03 04 05 06 07 08 09 0a 0b 0c 0d 0e 0f" and the cipher-text "69 c4 e0 d8 6a 7b 4 30 d8 cd b7 80 70 b4 c5 5a" are loaded into the input pins "keyI" and "initialState" as shown in Fig 6. The output pin "plainTextValue" holds the desired plaintext value "00 11 22 33 44 55 66 77 88 99 aa bb cc dd ee ff". Here the inputs and outputs are also verified according to the NIST publication.

Fig 7 shows the simulation result of the decryption process following the encryption process. In the normal mode input pin "decryptionFollowsEncryption" is set to one and the other five pins are set to zero so that the initial input key and the plaintext are loaded into the input pins "keyI" and "initialState" respectively. The output pin "cipherTextValue" holds the desired ciphertext value which is used as input to the decryption module. Finally, the desired plaintext is regained after decryption and displayed through the output pin "plainTextValue". As we know that the AES algorithm is a symmetric algorithm so that the same key is used for both the encryption and decryption process.

Fig 8 shows the simulation result of the encryption process in test mode. To test the encryption module using the expected number of pseudo-random patterns and deterministic patterns, two input pins, "bistMode" and "encryptionForRandom" are set to 1, and the other four input pins are set to 0. In this research, we have used forty pseudo-random test patterns and

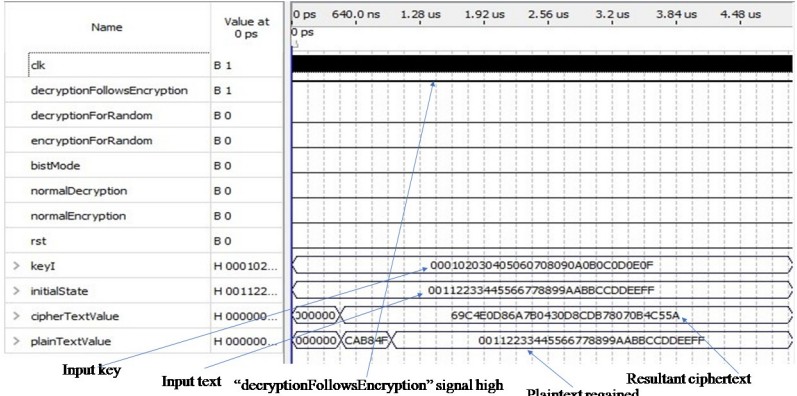

**Fig 7. Simulation result of decryption process following encryption process.**

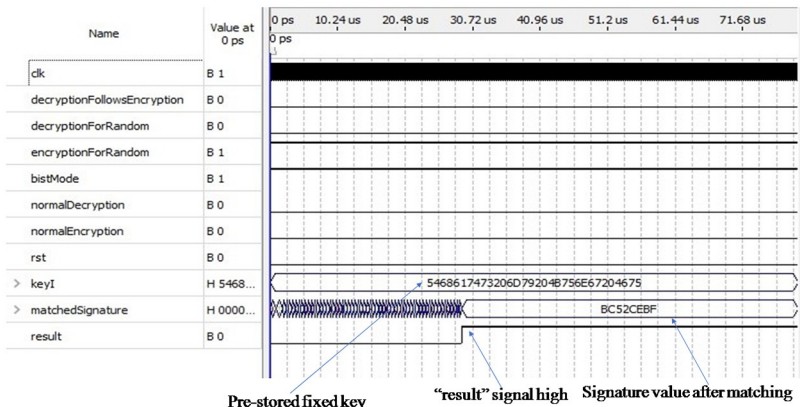

**Fig 8. Simulation result of encryption module in test mode.**

twenty one deterministic patterns. Sequentially, the expected number of pseudo-random patterns generated from the LFSR is fed as input to the encryption module. Consequently, we will get the expected number of ciphertext values; each of them is 128 bits in length which is also fed as input to the output response analyzer (ORA) sequentially. After that, 21 pre-stored partial seed values, each of them containing 10 bits, twenty-two zeros are appended in front of each seed value so to make each 32 bits in length are sent to the LFSR to form each of them into 128 bits in length and then fed to the ORA module one by one. The candidate signature which is obtained from the ORA module, is then compared with the golden signature "BC52CEBF" in a hexadecimal form stored in the memory. If there is a perfect match, then the output pin "result" is set to high, as shown in Fig 8 through the arrow sign to indicate successful testing; otherwise, it goes low to indicate failure. The output pin "matchedSignature" holds the signature value that perfectly matches the golden signature shown in the figure through the arrow shape. In test mode, we used a different pre-stored key "54 68 61 74 73 20 6D 79 20 4B 75 6E 67 20 46 75" than the key we used before in normal mode.

Fig 9 shows the simulation result of the decryption process in test mode. In this case, we used the same amount of pseudorandom and deterministic test patterns that we have used before to proceed with the test. Here, two input pins "bistMode" and "decryptionForRandom" are set to 1, and the other four pins are set to 0. The forty pseudo-random patterns

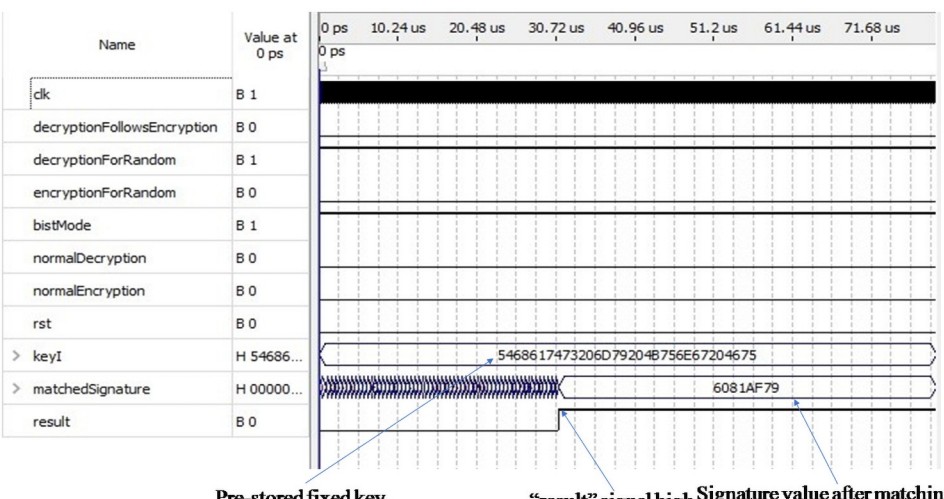

**Fig 9. Simulation result of decryption module in test mode.**

generated from the LFSR are fed as input to the decryption module one by one. Consequently, we will get forty ciphertext values; each of them is 128 bits in length which is also fed as input to the output response analyzer (ORA) one by one. After that, 21 pre-stored partial seed values, each of them containing 10 bits, twenty-two zeros are appended in front of each seed value so to make each 32 bits in length are sent to the LFSR to form each of them into 128 bits of length and then fed to the ORA module sequentially. The final response was obtained from the ORA module, which is our candidate signature compared with the golden signature "6081AF79". The "result" signal goes high after a perfect matching indicated by the arrow sign in Fig 9. The output pin "matchedSignature" holds the signature value after matching, which is also indicated by the arrow sign.

# 6 Comparison of results

Table 2 shows the comparison results of previous AES implementations using High-Level Language (HLL) on different platforms with this work. The parameters based on which the comparison is made with our result are platform, indicating different high-level implementation platforms, Data-path to indicate numbers of bits simultaneously processed by design, and BIST technique implementation. Table 2 shows that the proposed research is unique in terms

**Table 2. Comparison results of the AES in terms of BIST implementation.**

| Research | Platform | Data-path, BIST technique |
|---|---|---|
| Ali et al [42] | Quartus | 128, not used |
| Cao et al [43] | ModelSim | 128, not used |
| Yin et al [44] | ModelSim | 128, not used |
| Sever et al [45] | CMOS | 128, not used |
| Shastry et al [46] | Cadence NCSim | 128, not used |
| Chih et al [47] | CMOS | 128, not used |
| Liu et al [48] | CMOS | 128, not used |
| Bo et al [49] | CMOS | 128, not used |
| **This work** | Quartus | 128, Properly implemented |

of BIST implementation. We have implemented BIST successfully in our work, but none of the other researchers have implemented BIST. We didn't find any literature related to BIST implemented AES Crypto-processor ASIC.

## 7 Conclusion

AES outperforms all the others among the existing symmetric cryptographic algorithms, and a lot of applications are emerging based on this. Hardware implementation offers tremendous speed and impressive security than that of its software implementation. So a number of researches are proposed for the hardware implementation of AES. However, the testability problem is not addressed anywhere, which is now a burning issue for any complex VLSI chip. This research aims to overcome this limitation by designing the AES Crypto ASIC implemented with the mixed-mode BIST technique. The ASIC has been designed using Verilog HDL, which is now industry-standard software for VLSI design. The ASIC is simulated in its different mode of functionalities using two EDA tools: Modelsim and Quartus II. Simulation results in both environments prove the correctness of the design. Proper functionality has also been verified using NIST-provided inputs and outputs. The current research is compared with those of other researchers, and it shows that our research is unique in terms of BIST implementation.

## Supporting information

**S1 Appendix. Source code.** Controller module to all other modules are presented in the appendix.
(DOCX)

## Acknowledgments

The authors gratefully acknowledge to IICT, BUET for conducting this research using all kinds of facilities of the Advanced Embedded System Laboratory in the institute.

## Author Contributions

**Conceptualization:** Fakir Sharif Hossain.

**Data curation:** Md. Shazzatur Rahman, Fakir Sharif Hossain.

**Formal analysis:** Fakir Sharif Hossain.

**Investigation:** Md. Liakot Ali.

**Methodology:** Md. Liakot Ali, Md. Shazzatur Rahman.

**Resources:** Md. Liakot Ali.

**Software:** Md. Shazzatur Rahman.

**Supervision:** Md. Liakot Ali.

**Writing – original draft:** Fakir Sharif Hossain.

**Writing – review & editing:** Md. Liakot Ali.

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
