## [Decision Letter · Decision Letter 0]

5 Mar 2021

PONE-D-21-00858

Design of a built-in self-test implemented AES crypto-processor application specific integrated circuit

PLOS ONE

Dear Dr. Ali,

Thank you for submitting your manuscript to PLOS ONE. After careful consideration, we feel that it has merit but does not fully meet PLOS ONE’s publication criteria as it currently stands. Therefore, we invite you to submit a revised version of the manuscript that addresses the points raised during the review process.

We look forward to receiving your revised manuscript.

Kind regards,

Jun Ma, Dr.

Academic Editor

PLOS ONE

Reviewer #1: The authos present the article with ID PONE-D-21-00858, intitle “Design of a built-in self-test implemented AES crypto-processor application specific integrated circuit”, which may be of interest to the readers of this Journal, however, I have some concerns that I list below:

C1. Abstract. Author should follow the style of a structured abstract, which is based on the IMRAD structure of a paper, but without using headings . In other words, give a background and motivation to the paper, a brief description of the methods, the principle results, then conclusions or interpretations. In the particular case of this article, it is necessary to clarify the methods, main results and brief conclusions.

C2. The introduction is weak. In order to improve the quality of the manuscript and clarify its contribution, I suggest that the Introduction be updated with the trends of the latest generation of ciphers/cryptosystems using different methods and technologies, please discuss the following related work and mention the advantages of the proposed method in this manuscript.

DOI: 10.1002/cta.2759,

DOI: 10.1145/3199478.3199491,

DOI: 10.1007/978-3-642-20542-2_9,

DOI: 10.1016/j.chaos.2020.109646,

DOI: 10.1109/ICECS.2018.8617840

C3. It is suggested to add "code listing" of the main code fragment (Verilog HDL) of the proposed crypto-processor. Consider the main or relevant processes.

C4. It is recommended to add a Security Analysis, i.e., the well-known security analysis that are used to test and validate its robustness against different attacks.

C5. To clarify the contribution of this manuscript, it is suggested to add a comparative analysis of the main results obtained in the security analysis versus related work, for example, the authors can add a comparative table.

C6. It would have been interesting that the authors present a Halstead Complexity Analysis versus other related work.

C7. The results presented about the performance on FPGA do not show that the main findings improve the works reported in the state of the art, it is necessary to make a more detailed comparison of the performance versus some relevant works of the state of the art.

C8. The conclusions are not supported by the data.

C9. I suggest that all changes be highlighted in the manuscript. Please indicate point by point each of the responses to these comments and in which line of the manuscript it can be found.

After having carefully reviewed the manuscript and according to the quality standards of this Journal, for this reviewer it is not clear what is the novelty of this paper, what is its main contribution and advantages that it has versus the relevant works of the state of the art. Therefore, in my opinion, I consider that the manuscript is not yet ready for publication.

Finally, I hope that these suggestions help to improve the quality of the manuscript.

Reviewer #2: Pros:

1. Explanation of AES algorithms is precise and good

2. Explanation of AES crypto core is ok.

Cons:

1. In abstract it is mentioned that to solve the problem in BIST implementation. But neither problems are addressed nor implemented

2. Existing BIST methods are not addressed and surveyed for AES crypto processor

3. In introduction part, lack of continuity such as algorithms, hardware and software implementation and Test

4. There is no clear evidence to select the mixed mode BIST i.e .No reference not mentioned about the cons of the existing methods. There is no justification about the choice of Mixed mode BIST

5. This paper has lack of literature survey about testing of the VLSI circuit, self test, BIST and also pseudo random and deterministic techniques

6. Selection of Rom compression and LFSR reseeding is not clearly mentioned.

7. Comparison is not done properly; reference papers in table 2 are not relevant to the proposed work.

8. Paper organization is not up to the journal standard. Section is not clearly mentioned.

---

## [Author Response · Author response to Decision Letter 0]

16 Sep 2021

Response to Reviewers

Editor req-1. Please ensure that your manuscript meets PLOS ONE's style requirements, including those for file naming. The PLOS ONE style templates can be found at

Answer: Thank you so much for your comment. We have checked our manuscript through the PLOS ONE style templates and written in Latex of Plos One latex version 18.

Editor req-2. We suggest you thoroughly copyedit your manuscript for language usage, spelling, and grammar. If you do not know anyone who can help you do this, you may wish to consider employing a professional scientific editing service. 

 Answer: We copyedit our manuscript in peer and also checked by Prof. Dr. Muhammad. Sheikh Sadi, Department of Computer Science and Engineering, Khulna University of Engineering and Technology (KUET), Khulna, Bangladesh.

Editor req-3. In your Data Availability statement, you have not specified where the minimal data set underlying the results described in your manuscript can be found. PLOS defines a study's minimal data set as the underlying data used to reach the conclusions drawn in the manuscript and any additional data required to replicate the reported study findings in their entirety. All PLOS journals require that the minimal data set be made fully available. For more information about our data policy, please see http://journals.plos.org/plosone/s/data-availability.

Answer: All data are within the manuscript. We add an appendix of our source code in supporting information field. 

Reviewer #1: The authors present the article with ID PONE-D-21-00858, intitle “Design of a built-in self-test implemented AES crypto-processor application specific integrated circuit”, which may be of interest to the readers of this Journal, however, I have some concerns that I list below:

C1. Abstract. Author should follow the style of a structured abstract, which is based on the IMRAD structure of a paper, but without using headings. In other words, give a background and motivation to the paper, a brief description of the methods, the principle results, then conclusions or interpretations. In the particular case of this article, it is necessary to clarify the methods, main results and brief conclusions.

Answer: The abstract is rewritten as per suggestion of the honorable reviewer. Thanks for the suggestion.

C2. The introduction is weak. In order to improve the quality of the manuscript and clarify its contribution, I suggest that the Introduction be updated with the trends of the latest generation of ciphers/cryptosystems using different methods and technologies, please discuss the following related work and mention the advantages of the proposed method in this manuscript.

DOI: 10.1002/cta.2759,

DOI: 10.1145/3199478.3199491,

DOI: 10.1007/978-3-642-20542-2_9,

DOI: 10.1016/j.chaos.2020.109646,

DOI: 10.1109/ICECS.2018.8617840

Answer: As per suggestion of the honorable reviewer, the introduction of the manuscript has been modified and upgraded. The references suggested by the reviewer have been accommodated. However we would like to mention politely that the focus of the research is not cryptography algorithm at all, rather to address the testability issues of the AES crypto-processor chip. AES is a proven symmetric key and latest cryptography algorithm adopted by USA military. Moreover the research papers suggested by the reviewer are asymmetric key cryptography which is not suitable for bulk data transmission. So we could not accommodate this comment. 

C3. It is suggested to add "code listing" of the main code fragment (Verilog HDL) of the proposed crypto-processor. Consider the main or relevant processes.

Answer: As per suggestion of the honorable reviewer, we have included the code in the Appendix of the manuscript. Thanks for the suggestion.

C4. It is recommended to add a Security Analysis, i.e., the well-known security analysis that are used to test and validate its robustness against different attacks.

Answer: we would like to again mention politely that the focus of the research is not cryptography algorithm at all, rather to address the testability issues of the AES crypto-processor chip. So the security analysis and to find the robustness of the AES algorithm are not the main theme of the current research. It is a proven fact that different crypt-analytical attacks such as Brute-force, Linear crypt-analysis and Differential crypt-analysis, etc., have been proven ineffective to break the AES. We are sorry to say that we have not accommodated the comment.

C5. To clarify the contribution of this manuscript, it is suggested to add a comparative analysis of the main results obtained in the security analysis versus related work, for example, the authors can add a comparative table.

Answer: we would like to again mention politely that the focus of the research is not security analysis of the cryptography algorithms. It is a proven fact that AES outperforms all other existing symmetric key cryptography algorithms. We are sorry to say that we have not accommodated the comment.

C6. It would have been interesting that the authors present a Halstead Complexity Analysis versus other related work.

Answer: we would like to again mention politely that the focus of the research is not complexity analysis of the cryptography algorithms. It is a proven fact that AES outperforms all other existing symmetric key cryptography algorithms. We are sorry to say that we have not accommodated the comment.

C7. The results presented about the performance on FPGA do not show that the main findings improve the works reported in the state of the art, it is necessary to make a more detailed comparison of the performance versus some relevant works of the state of the art.

Answer: The contents of the result section have been upgraded. It is as per our Contribution as claimed in the introduction section. 

C8. The conclusions are not supported by the data.

Answer: The contents of the conclusion section have been upgraded. It is now as per our Contribution as claimed in the introduction section. 

C9. I suggest that all changes be highlighted in the manuscript. Please indicate point by point each of the responses to these comments and in which line of the manuscript it can be found.

After having carefully reviewed the manuscript and according to the quality standards of this Journal, for this reviewer it is not clear what is the novelty of this paper, what is its main contribution and advantages that it has versus the relevant works of the state of the art. Therefore, in my opinion, I consider that the manuscript is not yet ready for publication.

Finally, I hope that these suggestions help to improve the quality of the manuscript.

Answer: Thanks the reviewer for the careful examination of the manuscript. We would like to mention politely that appropriate comments have been accommodated as best as possible.

Reviewer #2: Pros:

1. Explanation of AES algorithms is precise and good

2. Explanation of AES crypto core is ok.

Cons:

1. In abstract it is mentioned that to solve the problem in BIST implementation. But neither problems are addressed nor implemented

Answer: Thanks the reviewer for the careful examination of the manuscript. We would like to mention politely that the texts written in the abstract are as follows:

“At today’s VLSI design, the testability of a complex VLSI chip is the prime concern. The

main purpose of this research is to address this problem by implementing a mixedmode BIST technique into the chip”

The above texts indicate that we would like to introduce the concept of BIST in designing the AES crypto-processor ASIC which is not reported in the literature yet. We have implemented the state of the art Mixed-mode BIST technique in the ASIC.

Sorry for misunderstanding. In the revised manuscript we have upgraded the text to avoid any confusion.

2. Existing BIST methods are not addressed and surveyed for AES crypto processor

Answer: Thanks for the comments. We have done a literature review for evolution of test technology of VLSI chip towards mixed-mode BIST in section 3.

3. In introduction part, lack of continuity such as algorithms, hardware and software implementation and Test

Answer: As per suggestion of the honorable reviewer, the introduction of the manuscript has been modified and upgraded. Lack of information continuity has been solved.

4. There is no clear evidence to select the mixed mode BIST i.e .No reference not mentioned about the cons of the existing methods. There is no justification about the choice of Mixed-mode BIST

Answer: Thanks for the comments. We have done a literature review for evolution of test technology of VLSI chip towards mixed-mode BIST in section 3. The objective of any BIST technique is getting higher number of fault-coverage using lower number of test vectors. Mixed-mode BIST technique is capable of providing highest fault coverage using lower number of test vector. This is a long ago established and popular BIST technique.

5. This paper has lack of literature survey about testing of the VLSI circuit, self test, BIST and also pseudo random and deterministic techniques

Answer: Thanks for the comments. We have done a literature review for evolution of test technology of VLSI chip towards mixed-mode BIST in section 3.

6. Selection of Rom compression and LFSR reseeding is not clearly mentioned.

Answer: Thanks for the comments. This is basically mixed-mode BIST technique. In this technique, the LFSR is initialized with a seed value at the beginning and a pre-defined number of pseudo-random test patterns are generated. Then the same LFSR is reseeded with predefined certain number of seed value to generate deterministic test patterns.

7. Comparison is not done properly; reference papers in table 2 are not relevant to the proposed work.

Answer: As per suggestion of the honorable reviewer, reference papers have been updated. Actually we want to mention politely that we want to draw a conclusion from the table 2 that none of any researcher has proposed testability issues of the AES crypto-processor chip. 

8. Paper organization is not up to the journal standard. Section is not clearly mentioned.

Answer: As per suggestion of the honorable reviewer, the feedback has been accommodated and section of the paper has been clearly mentioned.

---

## [Decision Letter · Decision Letter 1]

2 Nov 2021

Design of a BIST implemented AES crypto-processor ASIC

PONE-D-21-00858R1

Dear Dr. Ali,

We’re pleased to inform you that your manuscript has been judged scientifically suitable for publication and will be formally accepted for publication once it meets all outstanding technical requirements.

Kind regards,

Jun Ma, Dr.

Academic Editor

PLOS ONE

Additional Editor Comments (optional): 

Reviewer #1: The authors have satisfactorily addressed all the comments and suggestions of this reviewer. The article is interesting and its contribution to the state of the art is clear.

---

## [Editor Report · Acceptance letter]

5 Nov 2021

PONE-D-21-00858R1 

Design of a BIST implemented AES crypto-processor ASIC 

Dear Dr. Ali:

I'm pleased to inform you that your manuscript has been deemed suitable for publication in PLOS ONE. Congratulations! Your manuscript is now with our production department. 

Kind regards, 

on behalf of

Dr. and Pro. Jun Ma 

Academic Editor

PLOS ONE